# Assessing the Impact of Lockdown on Atmospheric Ozone Pollution Amid the First Half of 2020 in Shenyang, China

**DOI:** 10.3390/ijerph17239004

**Published:** 2020-12-03

**Authors:** Liyuan Wang, Ju Wang, Chunsheng Fang

**Affiliations:** College of New Energy and Environment, Jilin University, Changchun 130012, China; Liyuanw19@mails.jlu.edu.cn (L.W.); fangcs@jlu.edu.cn (C.F.)

**Keywords:** ozone, lockdown, pollution events, HYSPLIT, trajectory channel

## Abstract

During the eruption of COVID-19, a citywide lockdown was executed from 26 January to 23 March 2020, in Shenyang, in which the ozone pollution has recorded significant variations. This paper mainly anatomized the comprehensive characteristics and evolution trends of ozone pollution based on the lockdown period in the first half of 2020. Using the Hybrid Single Particle Lagrangian Integrated Trajectory (HYSPLIT) model and cluster analysis method to establish backward trajectories and channels, the spatial transport process of ozone in the preset period and the causation of typical ozone pollution events were investigated in depth. The results demonstrated that: The ozone concentration pollution in the first half of 2020 was increased than last year. During the lockdown period, the basic pollutants levels were lower than that in pre-lockdown under different proportions, except O_3_ maximum 8-h moving average (MDA8) was increased by 69.7%, accompanied by the delay of daily peak value, increased pollution days and longer pollution cycle. The typical pollution events were highly consistent with the evolution path of fine aerosol compelled by extreme weather. The ozone concentration and the atmospheric oxidation capacity can be stably maintained at a low level when NO_2_ concentration remained at 50–70 μg/m^3^, no matter how much the AQI was. Meanwhile, ozone concentration in the downwind suburban was as low as the central city and soared in few stations amid post-lockdown, simultaneous the correlation between ozone and other pollutants converted from negative to positive. The trajectory indicated that the pollution sources during the lockdown and pre-lockdown were basically Southern Russia, Inner Mongolia, and the three provinces of Northeast China, the pollution from the Bohai Sea provoked ozone pollutants in Shenyang to rebound briskly amid post-lockdown, the pollution of neighboring countries and areas would have a stronger impact on air quality under the effect of lockdown.

## 1. Introduction

The COVID-19 epidemic broke out on a global scale in 2020, and global health is undergoing an emergency. Nearly 90% of the countries around the world have imposed stringent social isolation measures against the pandemic. Huge economic losses and unprecedented economic catastrophe were aroused under the shutdown of global social activities [1]. For all that, the lockdown compelled by the epidemic had an unexpected impact on the environment. As the lockdown has led up to a meager level on various industrial production, vehicle movement and social activities for a long time, the environmental pollution intensity across countries dropped drastically just within a few days [2]. It is of the essence to comprehend the environmental self-renewal ability and whether the lockdown is an effectual alternative measure to be executed for restricting air pollution.

The lockdown caused by the epidemic had a positive impact on environmental air and water quality [3]. Decreases in NO_2_ levels over China during lockdown (10–25 February 2020) compared to pre-lockdown (1–20 January 2020) were recognized by satellite data from NASA and the European Space Agency [4]. Concentrations of CO and NO_2_ in Almaty, Kazakhstan reduced sharply by 49% and 35%, respectively, during the lockdown compared with the pre-lockdown, except O_3_ levels mounted by 15% [5]. Morocco discussed the variations of PM_10_, SO_2_ and NO_2_ on 2 March 2020, which decreased by 75%, 49% and 96%, respectively [6]. Bangladesh perceived the variations in PM_2.5_, PM_10_ and NO_2_ concentrations during the lockdown, resulting in a significant reduction of 40%, 32% and 13%, respectively [7]. Remarkably, among all the pollutants discussed, ozone concentration increased during the lockdown. For instance, the average daily ozone concentration in Nice, Rome, Turin, Valencia of southern Europe and Wuhan increased by 24%, 14%, 27%, 2.4% and 36% [8]. Another research showed that compared with the pre-lockdown period, the ozone concentration rocketed briskly with an increase of 116.6% in Wuhan [9]. The ongoing discussions on this phenomenon have emphasized that the increase was mainly due to the unprecedented reduction of NOx, which led to a decrease in the O_3_ titration of NO [10,11].

The Chinese government has adopted rapid and drastically defensive measures to block the spread of the COVID-19 epidemic. Countless provinces and cities across the country have successively initiated first-level responses to major public health emergencies [12], which significantly alleviated the spread of the epidemic (The National Health Commission, http://www.nhc.gov.cn/), and created good conditions for accelerating the resumption of employment and fabrication throughout the country [13]. In this unprecedented emergency, industrial production and resident lives have undergone major adjustments, as well as emissions fluctuations and air pollutants variations significantly. 2020 marks the end of “Three-year Action Plan for the Blue-Sky Defense”, ozone pollution is becoming a new environmental problem when China’s PM_2.5_ governance begins to improve significantly after years of efforts. From 2013 to 2017, the average ozone concentration in 74 major cities increased by 20% [14]. As the provincial capital of Liaoning Province and the economic center of Northeast China, Shenyang has ranked first in ozone pollution compared with other neighboring cities [15,16]. Judging by the 90th of O_3_-8 h, whose concentration in Changchun was 138 μg/m^3^ in 2017 [17], and 136 µg/m^3^ in 2018 in Harbin [18], which were both lower than that in Shenyang (162.5 µg/m^3^ in 2018, 166 µg/m^3^ in 2017) [19]. The Shenyang municipal government has made great efforts in ozone control, compiling the “Special emergency response measures of ozone in heavily polluted weather in Shenyang”, and focusing on the sources of key industries, service and traffic; however, it still had little effect. According to the content of the government work video conference held in early May 2020, due to the influence of straw burning and sand blowing weather in April, the ozone concentration in Shenyang deteriorated by 0.8% year to year. Consequently, this study quantifies the air quality variations magnetized by COVID-19 during the pre-lockdown, lockdown and post-lockdown, and considered the effects of long-distance transmission. Furthermore, the ozone pollution processes of four events in the first half of 2020 were especially estimated to take a scientific view of the air quality during the lockdown, simultaneously facilitated Shenyang to appropriately consummate pollution prevention and restraint work in the future based on one’s own characteristics.

## 2. Materials and Methods

### 2.1. Study Area

Liaoning Province is an important coastal province in the three Eastern Provinces of China. As the capital of Liaoning Province, Shenyang, located in the center of Northeast Asia and Bohai economic circle, along with the Liaodong Peninsula, is regarded as a comprehensive transportation hub connecting the Yangtze River Delta, Pearl River Delta and Beijing–Tianjin–Hebei to the Kanto region. The research on air pollution in Northeast China has been lacking in recent years, of which the ozone pollution in Shenyang was the most omitting related research covered from 2013 to 2017, methods included the characteristics and differences of basic concentration variations, the effects of meteorological elements and ozone precursors [20,21] and the pollution exposure effects on respiratory diseases and lung cancer mortality [22]. Considering the harsh ozone pollution in Shenyang, along with its special position, we set the study scope in the urban area of Shenyang, scientifically analyze and discuss the pollution situation, which is of great significance to reveal the pollution in typical areas.

### 2.2. Data Acquisition and Processing

This study obtained hourly data of Air quality index (AQI) and six kinds of basic pollutants of O_3_, PM_2.5_, PM_10_, NO_2_, SO_2_ and CO at 9 national controlled sampling stations from 1 January to 30 June 2020, from Shenyang Ecological Environment Monitoring Center of Liaoning Province, as well as hourly meteorological data from Shenyang Meteorological Observatory, in which O_3_ was treated by the sliding eight-hour average method, MDA8 represents the daily maximum eight-hour average ozone concentration specified in the Ambient Air Quality Standard (GB3095-2012). The nine national control stations include Cultural Road (CTR), Lingdong Street (LDS), Forest Avenue (FRA), Dongling Road (DLR), Riverside Road (RSR), Xinxiu Street (XXS), Jingshen Street (JSS), Hunnan East Avenue (HEA), and Taiyuan Street (TYS). The location information of the monitoring station is shown in Figure 1.

We used the comprehensive index method to make descriptive statistics on the ozone pollution concentration data of Shenyang in 2020, evaluate the total scale, average level, relative level and dispersion degree of pollutants, and summarize its comprehensive characteristics and evolution trend. Before the data analysis, the hourly concentration of pollutants was processed by 8-h moving average, and the daily and monthly mean values were obtained according to the processed values. The standard deviation analysis and inter-annual deviation analysis of the pollutant concentration were carried out. We applied the HYSPLIT model and established a transmission channel to analyze the special pollution events, which was also supplemented by the vorticity evolution at 500 mb sea level pressure and fine aerosol optical thickness in East Asia from Modern-Era Retrospective analysis for Research and Applications (MERRA-2 Greenbelt MD USA). All appendices are placed at the end of the text. Finally, we set three periods according to the epidemic lockdown order and made a comparative analysis of the data to determine whether the lockdown effect existed. The correlation analysis was used to reveal the correlation between meteorological factors and pollutants, and the Kriging interpolation method was used to analyze the spatial ozone distribution.

### 2.3. Trajectory Clustering and Transmission Channel Establishment

The HYSPLIT model used in the long-distance transmission research in this paper is developed by the National Oceanic and Atmospheric Administration (NOAA) with Air Resources Laboratory (ARL) [23], combined with the analysis data of 1° × 1° provided by Global Data Assimilation System (GDAS). The initial height of the trajectory simulation is 100 m from the ground, and the trajectory estimation time is 72 h. The stepwise cluster analysis (SCA) algorithm was used to cluster the backward trajectories of the three preset periods caused by the epidemic [24,25]. Trajectories clustering was used to repeatedly merge the trajectories with the closest spatial similarity according to the transmission direction and the speed of each airflow track. Four representative clustering trajectories were finally obtained in each period in the paper.

The calculation formula of SCA is as follows:(1)$$\begin{array}{c} D=\sqrt{\sum_{j=0}^{t}dj^{2}}\\ \mathrm{SPVAR}=\sum_{i=1}^{X}\sum_{j=0}^{t}D{ij}^{2}\\ \mathrm{TSV}=\sum \mathrm{SPVAR} \end{array}$$
where *D* is the distance between any two trajectories; *t* is the trajectory transmission time; *i* is the backward trajectory number; *j* is the stop point number; *x* is the number of trajectories in the cluster; *dj* is the spatial distance between the *j*th stop points of the two trajectories; *dij* is the spatial distance from the *j*th stop point in the *i*th backward trajectory to the corresponding stop point of the average trajectory; SPVAR is space variation of each group of trajectories; TSV is total space variation.

In addition, the trajectories of the four events were connected together in the order of first backward then forward, to form a complete transmission channel to determine the whole process path of the source and destination of the pollutants.

### 2.4. Research Period Setting

#### 2.4.1. Division of the Lockdown Period

According to the order of the Shenyang COVID-19 Pneumonia Epidemic Prevention and Control Headquarters (No. 1), Shenyang has initiated the first-level response to the COVID-19 epidemic from 0:00 on 26 January 2020. From this moment, the city officially closed the city. According to order No. 9, since 23 March 2020, Shenyang has officially unblocked, with fully resumed work and production, and normal living order has been restored. Consequently, the data obtained in the first half of 2020 were divided into three periods as follows: 1 to 25 January as pre-lockdown (Previous lockdown); 26 January to 23 March as lockdown; 24 March to 30 June as post-lockdown.

#### 2.4.2. Division of Ozone Pollution Events

We perceived four special stages of ozone pollution and marked them out in Figure 2 as typical ozone pollution events for in-depth discussion. Event 1 occurred on 28–29 January. Event 2 occurred on 11–12 February. Both occurred in the early stage of the epidemic lockdown and were rare high pollution events compared with the same period year–on–year. Event 3 occurred on 13–16 April. Event 4 occurred on 29 April to 2 May, which were two special ozone pollution incidents that occurred after the city was released from the epidemic.

## 3. Results

### 3.1. Analysis of Ozone Pollution in Shenyang in the First Half of 2020

Since the lockdown in the first half of 2020 was exceptional, we made descriptive statistics on ozone pollution in the past 2018–2019 years in Figure A1 when focused on the analysis of 2020. There were 41 days that the O_3_-MDA8 exceeded the second-level standard in 2018. The highest value appeared on 24 June, reaching 288 μg/m^3^, which was 29 μg/m^3^ higher than the maximum on 24 May in 2019. In 2019, the degree of ozone pollution decreased, with 33 days exceeding the standard, and the seasonal variation throughout the year weakened. Compared with 2019, the daily ozone concentration in the first half-year of 2020 grew with an average ratio of 8.9%, the average daily ozone MDA8 concentration (96.24 μg/m^3^) increased by 1.4% and 1.2% lower than that in 2018. As same as 2019, the ozone concentration in 2020 began to exceed the standard in late April (Figure 2), but the pollution was more serious compared with previous years, such as the two rare phenomena that the concentration exceeded the first-class standard before March and was regarded as typical ozone pollution event 1 and 2. It can be seen from the box length of the violin plot (Figure 3) that, except for April and June, the non-abnormal data of other months were relatively concentrated. The anomalies in April came from the marked events 3 and 4. When compared to January–March, the increased ozone in April–June (post lockdown period) was mainly attributed to the seasonal variation of background tropospheric ozone, which determines the ozone variability, especially in sites with relatively low primary pollution [26]. Moreover, the photochemical ozone production during the January–March period was quite low, so that the surface ozone variability during the winter lockdown period was essentially determined by the tropospheric and boundary layer ozone background levels in combination with the NO titration in urban and suburban sites, influenced by the NOx emissions. In general, the dispersion degree of the data in the last three months was relatively high, and the distribution was basically concentrated in 100 μg/m^3^. The extreme value of ozone concentration was low in May, with few days of high concentration and low weekly average change.

On the contrary, the extreme value in April and June was higher, whose pollution period was much longer. Overall, as shown in Figure 4, 19 days of the first half-year of 2020 exceeded the standard; the number of pollution days and the extreme pollution values increased obviously. Moreover, there was no significant weekend effect, but it would become increasingly obvious as months increase on account of the decreasing trends in O_3_ precursor emissions magnetized by the complete lockdown of the epidemic [27,28]. The daily series of the box plot also showed that the concentration of ozone increased with months, and the concentration increased sequentially before, during the lockdown and the post-lockdown.

High ozone concentration was related to air mass aging and a higher degree of photochemical oxidation [29,30]. Ox(O_3_+NO_2_) plays an intermediate role in O_3_ and NO_2_ [30]. Both have the ability to trigger oxidative stress, but little research involves their comprehensive oxidative ability. Since the measurements of NO_2_ and O_3_ are interrelated through the rapid titration reaction of NO with O_3_, producing NO_2_, especially at urban sites, the comprehensive oxidation capacity (Ox) of the atmosphere can be evaluated by observing O_3_ and NO_2_. As shown in Figure 2, Ox had a long period of meager value during lockdown (108.01 μg/m^3^), when the atmospheric oxidation capacity and ozone pollution remained at a meager level at this stage. Combined with the correlation results (Figure 5), this phenomenon may be related to the meteorological conditions (Table A1) such as high relative humidity (70.3%), low temperature (0.48 °C), and small temperature difference were conducive to “breaking” the inversion layer over the city. Second, the precipitation (0.12 mm) at this stage was relatively high, and the average wind speed (2.44 m/s) was also higher than the overall average, which can also be reflected in the higher visibility (17.98 km). Nevertheless, the more feasible reason was the combined effect of O_3_ and NO_2_, consistent with the results of other reports [31,32], where a negative correlation also existed between them. Correlation analysis showed that the hourly mean concentration of ozone had a weak negative correlation with AQI, NO_2_ and relative humidity, but strong positive correlations with temperature and Ox and the positive correlation between Ox and NO_2_ indicated that NO_2_ plays an important role in atmospheric oxidation.

Air quality index (AQI) is an air quality assessment standard issued by the Chinese government in March 2012 to protect public health. It is an important index used to describe the degree of air cleanliness or pollution and its impact on human health, which can assess the health effects of breathing polluted air for hours or days. As shown in Figure 6, when the AQI was 50–200, the high ozone concentration occurred, and the NO_2_ concentration was between 10 and 50 and 70–130 μg/m^3^. When NO_2_ concentration was 50–70 μg/m^3^, no matter how much was the AQI value the ozone concentration stayed at a normal or low level, but the extremely meager level only appeared when NO_2_ concentration and AQI were both huge. In the first half-year of 2020, the photochemical oxidant Ox had a high positive linear correlation with O_3_ and NO_2_; when NO_2_ concentration seriously exceeded the standard, the ozone concentration depressed, which was different from the previous conclusion that Shenyang belonged to the NOx control zone [33].

### 3.2. Analysis of Typical Ozone Pollution Events

In order to have a more detailed understanding of ozone pollution in the first half of 2020, we defined four ozone pollution periods as pollution events. Event 1 occurred during the long New year holiday. Affected by epidemic control instructions and festivals, residents needed to reserve materials in large quantities, resulting in frequent human activities and increasing man-made emissions. Event 2 arose when the majority of residents left their accommodation where they have stayed for more than two weeks and began to expand the scope of their activities. Both events 3 and 4 occurred amid post-lockdown, in which the ozone concentration had already climbed hastily. It can be clearly found that the relative humidity was relatively lower than the around value during the four events in Figure 7. Event 1 appeared in winter with low temperature and humidity during the Lunar New Year; ozone pollution was aggravated by many local precursor pollutant emissions; therefore, further discussion of external transportation is needed. The temperature of event 2 was abnormal that promptly climbed from −10 °C the day before to 10 °C at first, then quickly recovered to beyond −10 °C at the end of the event. The temperature variation (Figure 7) of events 3 and 4 in mid-early April was almost consistent with the trend of ozone variations year on year (Figure 2). Extreme high-temperature conditions aggravated ozone pollution, and the influencing factors of this anomaly need to be further discussed. Therefore, we established transmission channels by connecting the forward and backward 72-h trajectories of each day during four events to characterize the whole pollution paths of the source and whereabouts intuitively.

As shown in Figure 8, the trajectories we discussed at this stage were much longer; each channel went from west to east, first backward, then forward. The backward trajectories of the four events all came from the eastern hemisphere, a small part of the forward trajectories extended to the Western Hemisphere under the influence of strong winds, the longest one spanned 50 degrees of latitude. Event 1 ran north-south, started from southern Russia, cut into Shenyang from northeast China, eventually passed through the Yellow Sea and went south into the Pacific Ocean. The backward trajectory of event 2 covered Inner Mongolia, Beijing–Tianjin–Hebei and the Bohai Sea; the forward trajectory reached the Bering Sea to the farthest. Both events 3 and 4 occurred in April, of which the source of the trajectory was roughly the Bohai and Hebei regions, respectively, and went east into the Pacific Ocean with inconsistent trajectory channel. To comprehend the unique pollution process in western China, we downloaded the diurnal evolution of fine aerosol optical thickness in East Asia during these four periods (Figure A2), recognized that the ozone variation in the four events was not only consistent with the fine aerosol evolution process but also the trajectory channels underwent the high aerosol pollution area. Numerous reports perceived that the aerosol optical thickness held a great influence on ozone concentration [34]. Indeed we recognized that the evolution of the two was deeply consistent, which were probably related to the meteorological system.

Wind speed in western China was extremely fast, with a large span and transition, which led to the rapid blowing of pollutants to the Pacific region. Event 3 appeared on April 16 was quite special; the transmission channels formed a spiral path with a rotation of 360 degrees. Thus, we investigated the meteorological data of the day (Figure A3), recognized that the study area on this day was under the low-pressure anticyclone system, with relative eddy current intensity in the center of the cyclone as high as 28 × 10^−5^/sec, which was not conducive to the diffusion of pollutants and led to the accumulation of pollutants locally. Eventually, ozone pollution reached a peak in the first half of the year.

### 3.3. Evolution of the Ozone Pollution in Shenyang Affected by the Lockdown of the Epidemic

#### 3.3.1. Evolution of Ozone and Other Basic Pollutants during the Preset Periods

The epidemic lockdown was a turning point for the environment, and it is necessary to discuss the ozone pollution in stages and analyze the impact of the lockdown. Affected by the epidemic, the air quality throughout the country has increased in the meanwhile, among which the NO_2_, PM_10_, and PM_2.5_ have dropped significantly [35,36]. The national PM_2.5_ concentration dropped by 24.9% year–on–year, with the Yangtze River Delta stood ahead. Both ground and tropospheric NO_2_ concentrations were reduced by 20% to 30%, and the CO concentration was reduced by 17% year–on–year [13]. Figure 9 visualized the variations of pollutant concentrations in Shenyang amid the three preset periods. It was found that except for the continuous increase in ozone concentration (69.7%), the other five pollutants and AQI during the lockdown period decreased in varying degrees compared with the pre-lockdown period (NO_2_ decreased by 34.4%, AQI by 37.1%, PM_2.5_ by 46.4%, PM_10_ by 39.5%, SO_2_ by 45.6%, and CO by 43.3%, details were shown in Table A1). The ozone concentration amid post-lockdown increased by 60.4% compared with lockdown and 172.2% compared with the pre-lockdown. The hourly variations of O_3_ and NO_2_ in Figure 10 during the preset period were very striking, showing that the negative correlation between O_3_ and NO_2_ was not affected, and the peak value of ozone affected by titration still appeared after the valley value of NO_2_, regardless of whether there was a lockdown or not. What is different was that regardless of whether it was under lockdown control or not, the variations fluctuations of single-peak ozone and single-valley nitrogen dioxide throughout the day were delayed overall. Existing research concerning the ozone in Shenyang from 2013 to 2015 reported that its daily variation peaks generally appeared at 14:00–15:00 [37], while in this study, they were at 17:00–18:00. This delay was likely due to the significant reduction of ozone precursors and the broken synergy between VOCs and NOx during the lockdown. The concentration of surface ozone increased by 5% year on year during the epidemic, mainly decreased in the North China Plain and increased in the southeast [13]. Shenyang is located in northeastern China, and the prevailing wind direction was southwesterly in summer and post-lockdown. Excepting the complicated reasons for local emission variations, the increase in ozone concentration compared to previous years may also be associated with long-distance transmission.

The lockdown also caused an attractive alteration. As shown in the correlation visualization results in Figure 11, the correlation analysis results of the basic pollutants during the three preset periods clearly showed an abnormal phenomenon. Before the lockdown, ozone was negatively correlated with all pollutants discussed. Except for the weak correlation with AQI, ozone had a high level of negative correlation with the six basic pollutants. This result has also been demonstrated in other reports [38]. Amid the lockdown and post-lockdown period, the correlation between ozone and all the pollutants under discussion turned into a positive correlation. The relationship between ozone and other pollutants was generally complicated, but the correlation between ozone and NO_2_ was normally negative. The change of the correlation between them in this stage and the extremely low NO_2_ caused by the decrease of ozone titration reaction in a short time revealed the main reasons. The influence of external factors was not ruled out. During the lockdown, only SO_2_ was negatively correlated with ozone with an extremely weak correlation. The concentration of SO_2_ and CO did not rebound as quickly as other pollutants but continued to decline during post-lockdown. As the main pollutant in winter, SO_2_ was greatly affected by coal-burning during the heating stage; it was normal that the concentration continued to drop.

#### 3.3.2. Spatial Distinction and Long-Distance Transport of Ozone in Shenyang during the Preset Period

The extreme lockdown measures taken during the epidemic greatly cut off the pollution source emissions. Nonetheless, a simple analysis of the variations in the observed concentration of local pollutants was not enough. The pollutant concentrations of each monitoring station were spatially interpolated, as shown in Figure 12, it was discovered that the ozone spatial variations in three discussion periods were consistent with the temporal variation trend in the previous analysis, that the ozone concentration of each station was the minimum during the pre-lockdown and maximum amid post-lockdown, which was also higher than the average value in the first half of the year. The spatial distribution of ozone concentration can be clearly seen from Figure 12 that the ozone concentration in the northwest of Shenyang was all higher than that in the southeast region, of which the high value was distributed in the north of the urban area, the meager value was still concentrated in the center of the city. However, the spatial distribution of ozone in CTR station has increased sharply after unsealing, which may be due to the sudden increase in human activities in the dense residential group after the school opening, coupled with the full opening of subway lines, resulting in a sharp pollution increase within two months. During the lockdown and pre-lockdown of the epidemic, the dominant and high-speed wind direction in Shenyang was northerly, which became southwesterly in post-lockdown. Hence the ozone concentration was higher in the northeast than southwest before the lockdown and gradually higher in the northwest than in the southeast after unblocking. In this study, the ozone precursor pollutants dropped sharply during the pre-lockdown and lockdown period; the north wind with high speed may not only blow the pollutants to the downwind reaction area but also diluted the downwind pollutants. In addition to the increase in the overall pollution concentration, the spatial changes in pollutants during the 2020 lockdown period and the same period in 2019 are relatively small. In other words, the lockdown measures did not affect the spatial distribution of ozone concentration, which changed more after unsealing. Under the control of conventional pollution sources, the overall ozone concentration did not decrease but increased. Considering the tropospheric ozone influence on surface ozone concentration was expected to be important during the January–March period when the lockdown measures in Shenyang have taken place. As a major global factor in various regions, especially in the eastern Mediterranean during summer, but also in its western part during spring and many other regions [39,40,41], the vertical ozone transport in the troposphere influencing the boundary layer and surface ozone values [42,43]. Besides the back-trajectories are useful mainly for assessing the tropospheric ozone influence in combination with the fact that the synoptic meteorological conditions may lead to high tropospheric, boundary layer and surface ozone concentrations [44], it is very necessary to bring the analysis of long-distance transmission to clarify the underlying causes of ozone variation.

We used the HYSPLIT model to process the backward trajectories of the three preset periods, and four trajectories were clustered in each period to facilitate the source analysis of the pollution. As shown in Figure 13, the sources of the backward trajectories were relatively consistent before and during the lockdown. Pollutants were generally transmitted from the northwest, and long trajectories generally passed through southern Russia and cut into China from eastern Mongolia. Russia began to implement a comprehensive anti-epidemic policy of shutting down production and home isolation on 30 March 2020. Therefore, we can see in Figure 13 that when China stopped production completely to combat the COVID-19, the direction and transmission speed of the pollution trajectory from Russia had hardly changed compared with the same period in previous years, but the proportion of pollution during this period has increased. There was a small difference in climate change during the same period within two years when the pollution sources of the surrounding countries and regions of China had not changed significantly. The pollution in this period will have a greater impact on China that under lockdown along with the long-distance transmission, which was also reflected in Mongolia in northern China. Mongolia reported its first case of new coronary pneumonia on 2 March 2020, and announced a traffic blockade in the capital Ulaanbaatar and the central cities of the provinces until March 16. Comparing the changes in the trajectory during the blockade period in Figure 13, it is found that the long-distance pollution trajectory has shifted to Mongolia, and the trajectory has significantly changed the direction of transformation after the blockade is over. All these factors indicated that pollution in the surrounding areas had a negative impact on China’s air. The probabilities of short trajectories passing through Inner Mongolia, the Greater and Lesser Khingan Range as well as Horqin Sandy Land were also elevated. As shown in Table 1, the largest trajectory (30.67%, NO. 4) during the pre-lockdown came from North-northwest direction (NNW), started from the northern section of the Greater Khingan Mountains, along with the Northeast Plain, eventually passed through the three northeastern provinces (Western Heilongjiang Province, northwestern Jilin Province and Liaoning Province). The NO. 1 trajectory with the heaviest proportion during the lockdown is shown in Figure 13 and started from Huolinguole in Inner Mongolia, crossed the Songliao watershed and cut into the northwestern Liaoyang in Liaoning Province along the western Horqin Sandy Land, then affected the southwestern Shenyang, of which the ratio reached 40.28%, and the proportion was approximately 10% higher than the highest trajectory before the lockdown, indicating that the pollutants during the lockdown were greatly affected by the three northern provinces and the Horqin Sandy Land in the northeast. There were two special pollution trajectories, the first one was NO. 2 trajectory, accounting for 45.11%, which came from the Yellow Sea, rotated back to the Bohai Sea through a 180° rotation affecting by the atmosphere and cut into the Liaodong Peninsula, transmitted to Shenyang ultimately through the northeast Dalian, the southeast Anshan and Liaoyang, with the shortest length and the highest proportion, as well as the greatest impact on the study area. The second one came from the Ayano meisky economic zone in eastern Russia, along with Turana and Buleya Mountains China, across the Lesser Khingan Range and the northwest Heilongjiang, transmitted along the west side of the Northeast Plain to Shenyang. Based on the previous discussion, it was these two long trajectories that accounted for a relatively high proportion and were distinguished from the long trajectories from Russia and Mongolia in the previous two periods, which resulted in the rapid rebound of pollutants. Historical data demonstrated that the pollution level in Changchun was relatively high during the same period, and the pollution in Tongliao around Horqin Sandy Land was more severe. In Inner Mongolia, the first-level alert for epidemic prevention and control was sounded in early January, and the level was adjusted to third-level at the end of February. The rapid resumption of production and the resulting air pollution would have an adverse impact on the local and surrounding areas. Additionally, the high wind speed during the lockdown (2.2 m/s for average, 9 m/s for maximum) greatly promoted trans-regional transmission of pollutants.

## 4. Discussion

Throughout the first half of 2020, the overall fluctuation of ozone pollution has increased. On one hand, adverse meteorological conditions led to high ozone concentration, which was further confirmed by the analysis of four special ozone pollution events in the first half of the year. In the case of high humidity, the free Hydrogen and hydroxyl radicals contained in water vapor quickly decomposed ozone into oxygen molecules and reduced the ozone concentration, causing a negative correlation between ozone and humidity [45,46]. Surface ozone concentration increase with rising temperature and the chemistry variations attributed to temperature-driven increases in emissions of NOx are key drivers of increased ozone concentrations on hotter days [47,48]. High wind speed can raise the height of the atmospheric boundary layer and promote the transport of high mass concentration of O_3_ to the ground, resulting in the increase of surface O_3_ concentration. At the same time, it can also aggravate the horizontal diffusion, dilute the high O_3_ concentration, and greatly improve the pollution of ozone.

On the other hand, precursor pollutants also play an indispensable role. Numerous studies have shown that a remarkable reduction in PM_2.5_ may bring the potential risk of increased ozone content. When the concentration of PM_2.5_ in the atmosphere decreased significantly, it would lead to an increase of light radiation, which was beneficial to the formation of ozone. At the same time, the inhibition of particles on the photochemical process results in the decrease of atmospheric oxidation ability, which was the possible reason for the decrease of near-surface ozone concentration in 2020 [49]. China has effectively controlled PM_2.5_ while failing to regulate the proportion of NOx and VOCs, ultimately aggravating ozone pollution. Compared with previous years, the overall ambient air quality was further improved in 2018–2019; government work has achieved breakthrough results; nevertheless, ozone pollution became protruding. Furthermore, in the first quarter of 2020, the domestic ozone concentration increased by 3.4% from year to year, and 8.1% in April, while PM_2.5_ concentration fell by about 15% year–on–year. Research showed that O_3_-MDA8 continued to grow at an average rate of 4.6% every year [50], and Liaoning Province was already the province with the heaviest ozone pollution in Northeast China.

There was also sufficient and robust evidence that the short-term exposure to O_3_, NO_2_, particulate matter was positively correlated with ozone all-cause mortality [51,52], the death rate attributed to O_3_ was the highest proportion of the six basic pollutants, and the meager ozone pollution would cause higher mortality rate [53]. The chemical interconversion between NO_2_ and O_3_, coupled with the fact that both are associated with health effects, has also led to recent interest [54]. In our research, we found that NO_2_ concentration has a very important effect on ozone. The ozone concentration was the highest when the NO_2_ concentration was low, which also verified that the reduction of NO titration led to less ozone consumed and accumulated NO_2_. When AQI and NO_2_ concentrations in Shenyang reached standards, not only the atmospheric oxidation capacity turns into the weakest, but also the ozone concentration was in a steady-state. Most importantly, when NO_2_ reached 50–70μg/m^3^, ozone pollution would not affected by the comprehensive assessment index of air pollution. This characteristic proved the importance of coordinated ozone treatment, and the judgment of oxidation capacity could assist in determining the emission reduction ratio of ozone precursors in the future.

Similar to much other literature on research results of pollutant variations from the perspective of epidemic lockdown [55,56], during the lockdown period in Shenyang, the concentrations of various pollutants decreased to varying degrees, with the largest decline of PM_2.5_ (46.4%), NO_2_ has the smallest drop, but also reached 34.4%. On the contrary, the ozone concentration increased continuously during this period, with an increase of 69.7%. This study also found that the correlation between ozone and other pollutants was reversed during and after the strict implementation of the lockdown measures; ultimately, the reason was that the ozone titration effect was greatly reduced. Affected by the epidemic, the spatial distribution of ozone pollution in Shenyang in 2020 changed in different periods but was consistent with the same period in previous years. Shenyang has always followed the regular pattern that the concentration in the central urban area was lower than that in the suburbs [57]. In 2020, the spatial distribution of ozone in Shenyang changed slightly before and after the lockdown, and the concentration of some stations increased sharply after the closure. The tropospheric ozone and its subsequent influence, including tropospheric folding and subsidence on the boundary layer and surface ozone concentrations, had significant impacts on regional ozone transportation.

During the period of the epidemic, the lockdown measures greatly improved air pollution, but ozone pollution was more serious than before, and the pollution concentration was higher than that in previous years. After the unsealing of the epidemic, concentrations of various pollutants in the study area rebounded swiftly, and the long-distance transmission channel also made the pollution sources in Shenyang more complex. Considering the factors discussed comprehensively, the higher ozone levels in the lockdown period was not only caused by the long-distance transmission and simple increase of ozone concentration but also due to the reduced titration from reduced nitric oxide, and less ozone was consumed.

## 5. Conclusions

A multifaceted evaluation of the current ozone pollution in Shenyang revealed the impact of long-term lockdown on the environmental situation. Overall, the lockdown did not have a positive effect on ozone pollution, which still proliferated constantly. The results of the corresponding period analysis showed that the ozone concentrations during the lockdown in 2020 were 1.4% higher than that in 2019 and 1.2% lower than that in 2018. In general, compared with last year, ozone pollution was more serious in the first half of 2020. Comparing the variations of pollutants during the lockdown with the pre-lockdown period, it came across that the average concentration of AQI decreased by 37.1%, so did NO_2_ (−34.4%), PM_2.5_ (−46.4%), PM_10_ (−39.5%), SO_2_ (−45.6%) and CO (−43.3%) level, while O_3_-MDA8 increased by 60.4% amid post-lockdown than lockdown period, and 172.2% than pre-lockdown period. Moreover, the peak time of the daily variations of ozone and nitrogen dioxide was delayed, and the correlation between ozone and other pollutants transformed from negative to positive during the lockdown period. Of paramount importance, it was found for the primary time that no matter how much was the AQI value, the ozone concentration stably maintained at the standard level when NO_2_ concentration stained at a low level (50–70 μg/m^3^), which confirmed the prominence of coordinated ozone treatment. As for the four pollution events, we established the transmission channel by HYSPLIT to characterize the whole pollution paths of the source and whereabouts intuitively and found out that they were highly consistent with the evolution path regarding the high pollution region of fine aerosol compelled by high wind speed and low-pressure anticyclone involving by abnormal meteorological systems. Meanwhile, compared with the previous years, the spatial distribution of ozone pollution changed little. The ozone concentration in the northwest of Shenyang was all higher than that in the southeast region, of which the high value was distributed in the north of the urban area, the concentration in the downwind suburban was as low as the central city. High temperature and low humidity were still the main reasons for the overall increase in ozone temperature during post-lockdown. The abnormal phenomena of some stations, such as CTR, were related to the increase of traffic emissions and human activities. As the interaction between pollutants was very complex, the rapid increase in ozone pollution was even more related to the motion of the atmosphere. The HYSPLIT results demonstrated that the pollution source was consistent before and during the lockdown, and the trajectory basically consists of the long-distance transmission in southern Russia, dust impact in Inner Mongolia and short-range input from the three eastern provinces. The pollutants from the transmission channels in eastern Russia and the Bohai Sea after unsealing was one of the reasons for the hasty rebound of all pollutants. The backward trajectory analysis demonstrated that because of the different times and specific measures of lockdown, the pollution of neighboring countries, including Russia and Mongolia and the surrounding areas of Shenyang, would have a stronger impact on air quality under the effect of long-distance transmission.

This study corroborates the complexity of air pollution in Shenyang, and there is an urgent to further conduct the source distribution of ozone as well as the deep relationship of ozone precursors. The COVID-19 lockdown is an excellent opportunity to test environmental parameters and improve air quality. We should not only perceive the effect of short-term lockdown as an alternative policy measure to depress pollution, the economic impact and the cost-effectiveness of decision-makers in determining such controls also need to be perceived to accomplish systematic environmental control measures.

## Figures and Tables

**Figure 1 ijerph-17-09004-f001:**
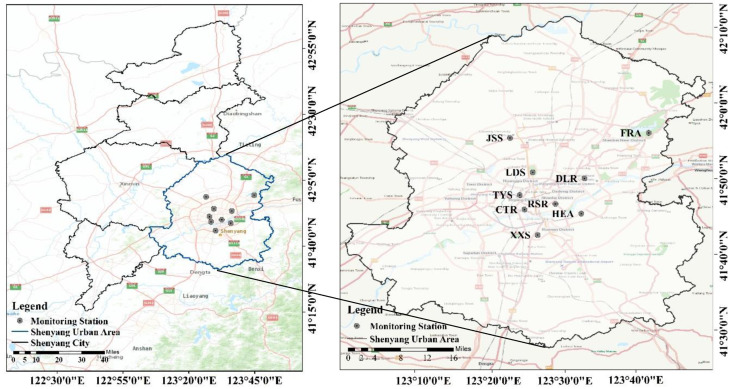
Geographical location of Shenyang with its urban area and monitoring stations.

**Figure 2 ijerph-17-09004-f002:**
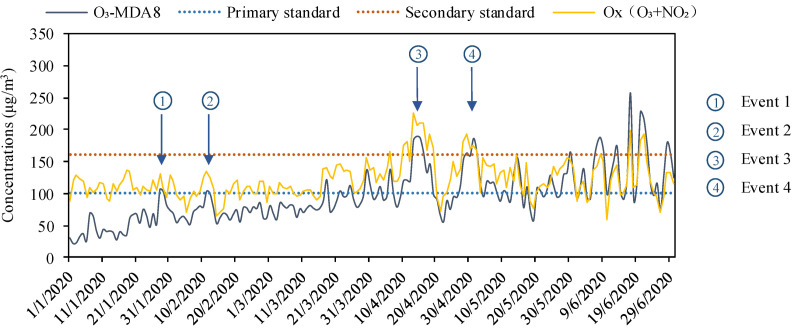
Diurnal variation of O_3_ maximum 8-h moving average (MDA8) and Ox with event tagging from January to June 2020.

**Figure 3 ijerph-17-09004-f003:**
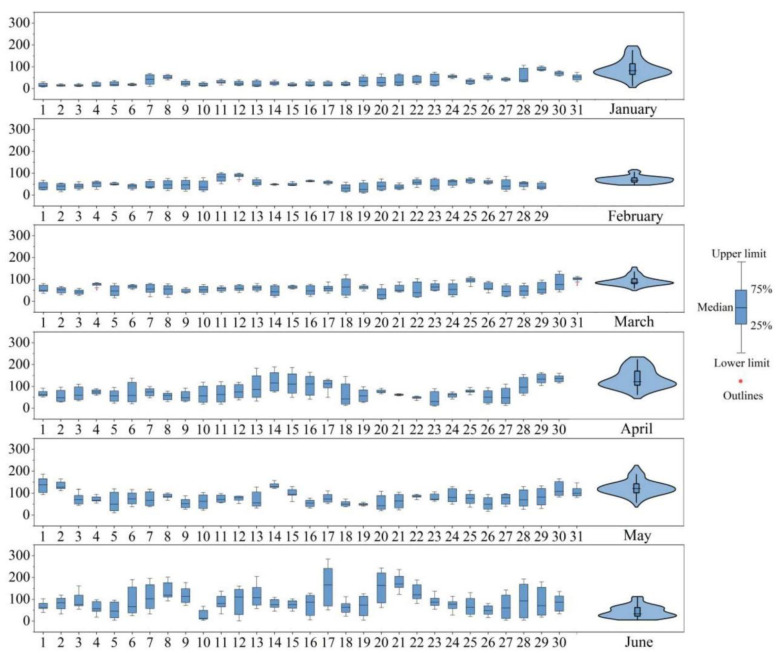
Daily series box plot and monthly average violin plot of ozone concentration from January to June 2020.

**Figure 4 ijerph-17-09004-f004:**
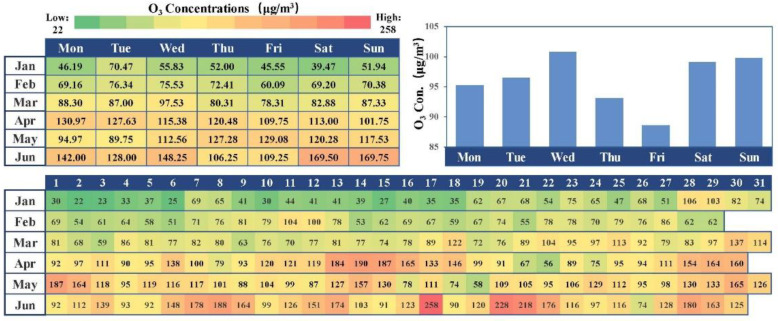
Status of daily average O_3_ concentration and weekly statistics in the first half of 2020.

**Figure 5 ijerph-17-09004-f005:**
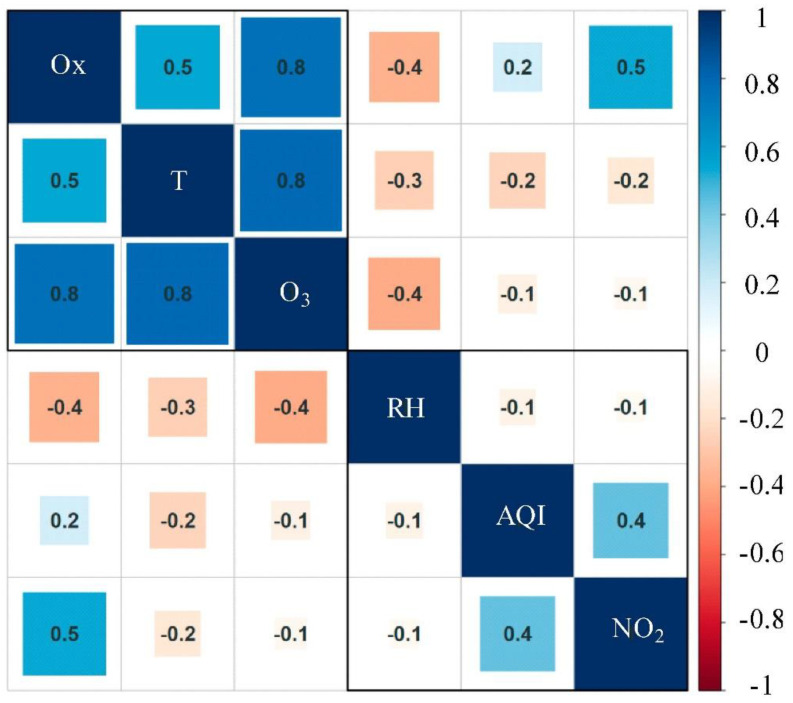
Correlations between pollutants and meteorological elements.

**Figure 6 ijerph-17-09004-f006:**
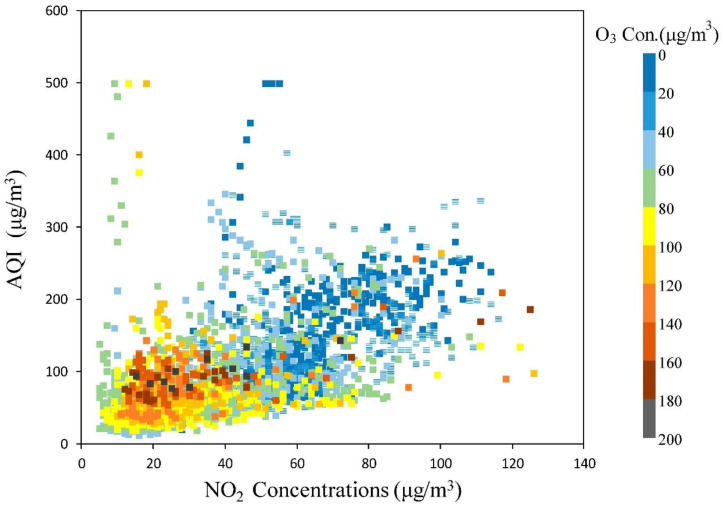
AQI and NO_2_ concentrations scatter chart (Ozone concentration as color information) from January to June 2020.

**Figure 7 ijerph-17-09004-f007:**
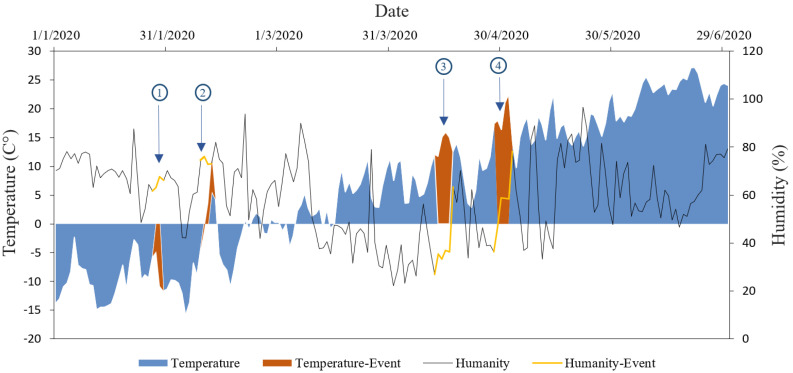
Diurnal variation of temperature and relative humidity (The four segments from left to right are events 1–4).

**Figure 8 ijerph-17-09004-f008:**
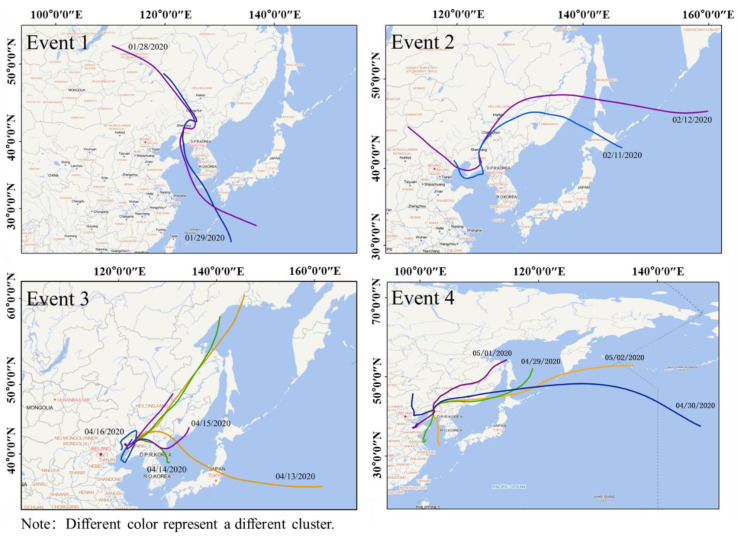
Transmission channels of pollution events 1–4 in Shenyang in the first half of 2020.

**Figure 9 ijerph-17-09004-f009:**
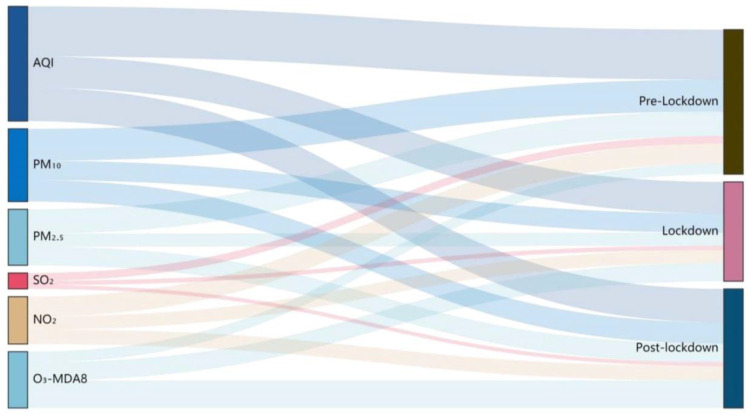
Concentration variations of air quality index (AQI) and five basic pollutants during the preset period.

**Figure 10 ijerph-17-09004-f010:**
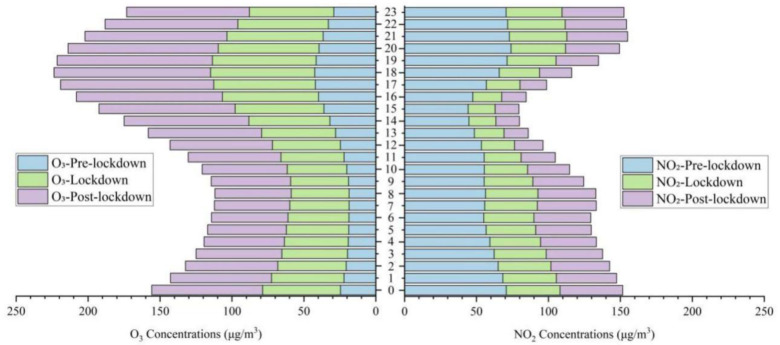
Hourly variations of O_3_ and NO_2_ concentrations during the preset periods.

**Figure 11 ijerph-17-09004-f011:**
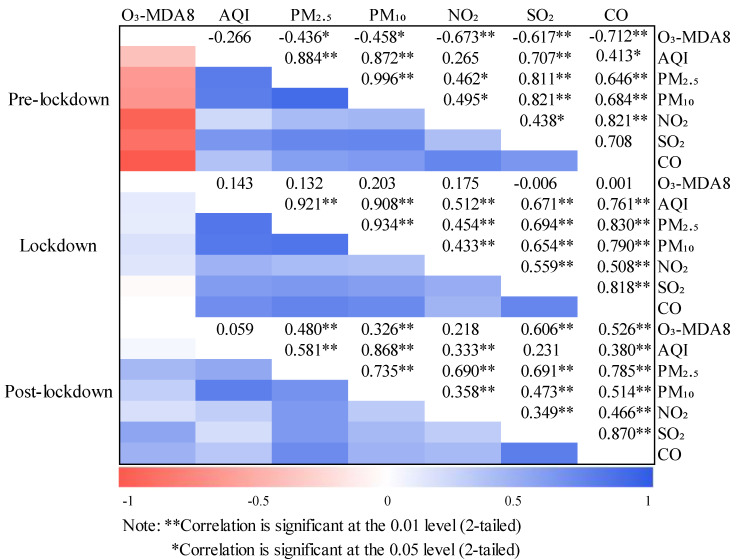
Correlation results between AQI and six basic pollutants in three preset periods.

**Figure 12 ijerph-17-09004-f012:**
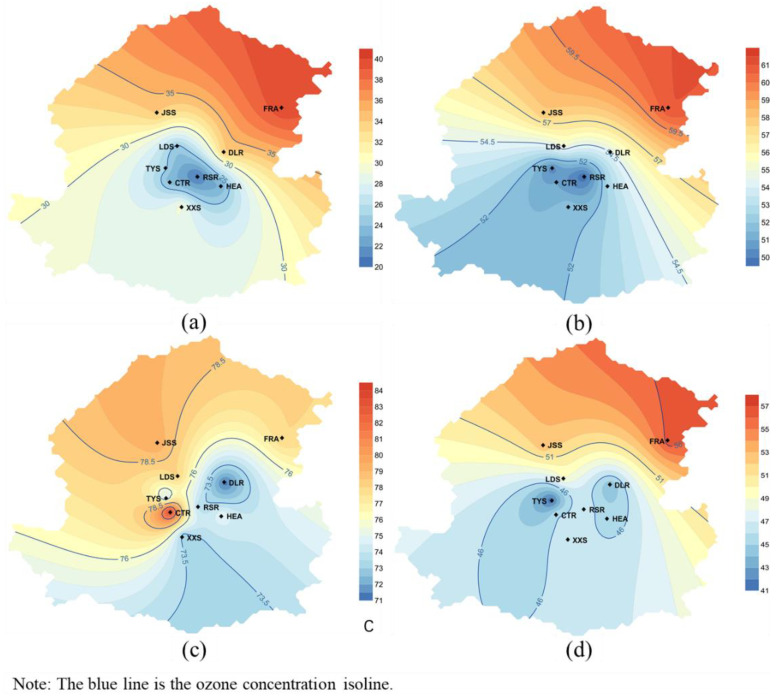
Spatial distribution of ozone pollution in Shenyang. (**a**) Pre-lockdown; (**b**) 2020 lockdown; (**c**) post-lockdown; (**d**) 2019 lockdown.

**Figure 13 ijerph-17-09004-f013:**
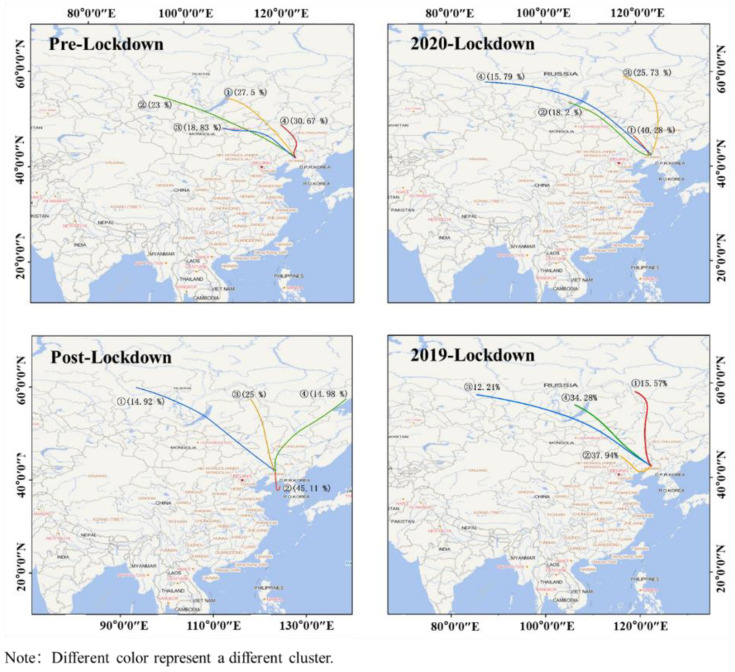
The backward trajectory clustering paths during the preset research periods.

**Table 1 ijerph-17-09004-t001:** Basic information of the clustering trajectory during the preset research periods.

Period	Cluster NO.	Direction	Areas of Pathways	Percentage of Total Trajectory (%)	Trajectory Length (km)
Pre lockdown	1	NW	Lake Baikal, Yabnorov and Borshov Mountains, Eastern Mongolia, Inner Mongolia Plateau, Greater Khingan Range, Horqin Sandy Land	27.50	1857.13
	2	WNW	East Sayan Mountain in Russia, Northeast Kent Mountain in Mongolia, Yanshan Mountains	23.00	2862.09
	3	WNW	Kent Mountain, Inner Mongolia Plateau, Horqin Sandy Land	18.83	1439.85
	4	NNW	Northern Greater Khingan Range, Northeast Plain, Western Heilongjiang, Northwest Jilin, Northern Liaoning	30.67	808.62
2020 lockdown	1	NW	Huolingole in Inner Mongolia, Songliao Watershed, Northwest Liaoyang, Western Horqin Sandy Land, Southwest Shenyang	40.28	569.22
	2	WNW	Yabnorov and Borshov Mountains, Northeast Mongolia, Northeast Xilinguole League in Inner Mongolia, Chifeng City, Southwest Fuxin, Northeast Jinzhou	18.20	1962.43
	3	NNW	Outer Hinggan Mountains in Russia, Middle Greater and Lesser Khingan Range, West Heilongjiang Province, Central Jilin Province, Northeast Plain	25.73	1973.17
	4	WNW	South Russia, Lake Baikal, Eastern Mongolia, Southwest Hulunbeir, Northeast Xilinguole League, Central Tongliao, Northwest Shenyang	15.79	3489.30
Post lockdown	1	WNW	Eastern Mongolia, Lake Baikal, Southwest Hulunbeir, Northeast Xilinguole League, central Tongliao, Northwest Shenyang	14.92	3150.11
	2	S	Yellow Sea, Bohai Sea, Liaodong Peninsula, Northeast Dalian, Southeast Anshan, Liaoyang	45.11	355.11
	3	NNW	South Russia, Great Khingan Mountains, Northeast Plain, Horqin Desert	25.00	1766.78
	4	NE	Ayano meisky, Turana and Buleya Mountains, Lesser Khingan Range, Northwest Heilongjiang Province, West side of the Northeast Plain	14.98	2114.64
2019 lockdown	1	N	Southeast Russia, Northeast Inner Mongolia, Northeast Plain, Tongliao	15.57	1772.01
	2	WNW	Xilin Gol League, Chifeng, Huludao, Liaodong Bay, Panjin	37.94	843.95
	3	WNW	South Russia, East Sayan Mountain, Lake Baikal, Yabnorov and Borshov Mountains, East Mongolia, Northeast Inner Mongolia, Horqin Sandy Land	12.21	4177.31
	4	NW	Lake Baikal, Yabnorov and Borshov mountains, East Mongolia, Northeast Inner Mongolia, Horqin Sandy Land	34.28	2150.36

Note: E for East; S for South; W for West; N for North.

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
