# Peer review of "Assessing the Impact of Lockdown on Atmospheric Ozone Pollution Amid the First Half of 2020 in Shenyang, China"

_ijerph, 2020, doi:10.3390/ijerph17239004_

Round 1
Reviewer 1 Report
The article presents changes in the concentrations of selected air pollutants in one of the Chinese cities during the lockdown in the first half of 2020, compared to the situation before and after lockdown. The results of calculating the backward trajectory paths are presented in an interesting way. Generally, I think it is a quite relevant field and the work can be published.
Strengths:
- extensive and very up-to-date literature,
- very clear graphic part of the research results.
Weaknesses:
- of course, at the stage of the final edition of the work: it is worth organizing the space and results so that they are easier to read (e.g. table 2 should be on one page, or at least it should also have a header on the next one; also Fig. 4 "detached" from the description);
- I have also doubts about the sentence in the conclusions, line 512: "In general, ozone pollution was more serious in the first half of 2020." Is this a comparison to the second half of 2020? Where are the results then?
Author Response
Our Revision:
First of all, thank you very much for your affirmation, which will give us great encouragement. Secondly, we are extremely appreciative of your review work and suggestions. We will provide a point-by-point response below.
Point 1: of course, at the stage of the final edition of the work: it is worth organizing the space and results so that they are easier to read (e.g. table 2 should be on one page, or at least it should also have a header on the next one; also Fig. 4 "detached" from the description)
Response 1: Thank you very much for your suggestion. The presentation of Table 2 is indeed not rigorous. We put Table 2 on one page. Figure 4 "detached" from the description because we expressed it incorrectly within the text. We have made adjustments in the revised manuscript L 193-194. "Overall, as shown in Figure 4, 19 days of the first half year of 2020 exceeded the standard, the number of pollution days and the pollution extreme values increased obviously." The 19 excessive days in the text are based on Figure 4, in the lower half of the figure, there are a total of 182 days of average daily ozone concentration values for six months. The grids are red and the one exceeding the standard (160μg/m3) are marked as one day. The upper part of the chart shows the weekly variation of ozone concentration per month and the first half of 2020, and detailed description is reflected in the original text. The text description in Figure 4 is more accurate after the modification according to your suggestion.
Point 2: I have also doubts about the sentence in the conclusions, line 512: "In general, ozone pollution was more serious in the first half of 2020." Is this a comparison to the second half of 2020? Where are the results then?
Response 2: Thank you very much for your suggestion. The expression of this sentence was defective, we have made adjustments in the revised manuscript of the conclusion part where this sentence is located in L 491-494. "The results of the corresponding period analysis showed that the ozone concentrations during lockdown in 2020 were 1.4% higher than that in 2019 and 1.2% lower than that in 2018. In general, compared with last year, ozone pollution was more serious in the first half of 2020." This conclusive sentence has been expressed in both the results and the discussion sections. The analysis of the original text considered that compared with 2019, the daily ozone concentration in the first half year of 2020 grew with average ratio of 8.9%, the average daily ozone MDA8 concentration (96.24μg/m3) increased by 1.4%. As same as 2019, the ozone concentration in 2020 began to exceed the standard at late April, but the pollution was more serious compared with previous years. Therefore, ozone pollution in the first half of 2020 was more serious than in the same period of 2019. The meaning of this sentence is more complete after the modification.
Submission Date
10 November 2020
Date of this review
23 Nov 2020 08:28:24

Reviewer 2 Report
This manuscript analyzes the concentrations of some pollutants, both during the confinement of the CODIV-19 pandemic, as well as before and after. The article is a significant contribution, both in the tools it uses to evaluate different events, and in the conclusions. In particular the behavior of ozone. Both the methodology and the discussion seem relevant to me. Likewise, the figures and tables are concise and are related to the main text. As a suggestion to the authors, I would like to suggest the following changes or clarifications:
- Place as Figure 8 was obtained.
- Indicate that Figures A1, A2, and Tables A1, A2 are at the end of the text.
- Figures 9 and 10 are difficult for me to understand, since they are counting the concentration of all pollutants in different periods.
- In line 471, it indicates “The analysis results from the perspective of epidemic lockdown were similar to lots of literature”.
Considering the above, I suggest that the changes noted that I hope will improve this interesting study be made before being published.
Author Response
Our Revision:
Thanks very much for your review work and suggestions. Your affirmation gives us great encouragement. We will provide a point-by-point response below.
Point 1: Place as Figure 8 was obtained.
Response 1: Thank you very much for your suggestion. It is our mistake in drawing the map. We have marked the location of Shenyang with black five pointed stars in Figure 8.
Point 2: Indicate that Figures A1, A2, and Tables A1, A2 are at the end of the text.
Response 2: Thank you very much for your suggestion. In order to avoid repeated explanation after each figure or table, we have explained the position of all appendixes in the manuscript L116-117. "All appendices are placed at the end of the text."
Point 3: Figures 9 and 10 are difficult for me to understand, since they are counting the concentration of all pollutants in different periods.
Response 3: Thank you very much for your suggestion. Figure 9 is a Sankey diagram of all pollutants in three periods. There are six pollutants listed on the left, three periods are on the right column, and each pollutant has three lines that merge into three periods. The width of the line represents the quantity of the pollutant entering the period. Comparing the width of lines of each pollutant flowing into the three periods, we can find the conclusion in the manuscript: "Except for the continuous increase in ozone concentration, the other five pollutants and AQI during the lockdown period decreased in varying degrees compared with the pre-lockdown period." Figure 10 is a stacked histogram of average hourly O3 and NO2 in the three periods. Since NO2 is an important precursor of O3, their hourly variations usually present the opposite trend. Generally speaking, O3 varies in a single-peak type, and NO2 varies in a single-valley type, and the time of the extreme value is close. Because of the special period of epidemic lockdown, it is necessary to analyze the hourly average concentration of these two pollutants. The abscissa of Figure 10 is the concentration, and the ordinate is 24 hours. The three colors represent three different periods. On the left is the hourly variation of O3, whose peak value is 18:00-19:00, the right part is hourly variation of NO2, and the valley value is 14:00-15:00, showing that the regular pattern between O3 and NO2 was not affected.
Point 4: In line 471, it indicates “The analysis results from the perspective of epidemic lockdown were similar to lots of literature”.
Response 4: Thank you very much for your suggestion. We revised this sentence in the manuscript to make its context more fluent. At the same time, two documents are cited to support this conclusion in L465-466. "Similar to lots of other literature on research results of pollutant variations from the perspective of epidemic lockdown [55,56], during the lockdown period in Shenyang, the concentrations of various pollutants decreased to varying degrees, with the largest decline of PM2.5 (46.4%), NO2 has the smallest drop but also reached 34.4%. On the contrary, the ozone concentration increased continuously during this period, with an increase of 69.7%."
[55] Wang, J.; Xu, X.; Wang, S.; He, S.; He, P. Heterogeneous effects of COVID-19 lockdown measures on air quality in Northern China. Appl. Energy. 2021, 282,116179. https://doi.org/10.1016/j.apenergy.2020.116179.
[56] Nie, D.; Shen, F.; Wang, J.; Ma, X.; Li, Z.; Ge, P.; Ou, Y.; Jiang, Y.; Chen, M.; Chen, M.; Wang, T.; Ge, X. Changes of air quality and its associated health and economic burden in 31 provincial capital cities in China during COVID-19 pandemic. Atmos. Res. 2021, 249, 105328. https://doi.org/10.1016/j.atmosres.2020.105328.
Submission Date
10 November 2020
Date of this review
15 Nov 2020 15:55:43

Reviewer 3 Report
Overview
The paper deals with the impacts on air pollution of COVID-19 lockdown measures during the first semester of 2020 in the city of Shenyang, China. There are some interesting results presented in this study and I think that the paper could be published, in principle, but in my opinion the authors need to deal with the following issues before publication:
General commends
At first, I think that the paper needs a thorough English language check as on many occasions the sentences, including the used terminology are not clear and straightforward enough. I would also suggest a reduction of the text, which should be focused on the description of the main results.
Also, when dealing with ozone pollution the quantity Ox=NO2+O3, called also “potential ozone” should be also examined (Kley et al., 1994). Although there is reference to that in Fig. 2, this is not repeated in the examination of the diurnal variations of ozone and NO2 (Fig. 10), where it should be clearly mentioned that the measurements of NO2 and O3 are interrelated through the rapid reaction of NO with ozone, producing NO2, especially at urban sites (NO titration).
An essential point when examining the variability of ozone concentrations in Fig. 3, the increased ozone in March-April (post lockdown period), when compared to January – February (lockdown period) should be mainly attributed to the seasonal variation of background tropospheric ozone, which determines the ozone variability especially in sites with relatively low primary pollution. In general, ozone rural background levels during spring are significantly higher than during winter. In addition to the above, I would suggest that when referring to the regional ozone transportation, the issue of tropospheric ozone and its subsequent influence on the boundary layer and surface ozone concentrations should be also considered. In relation to that, in my opinion, a weak point of the paper is that the levels of measured surface ozone are mainly related (or attributed) exclusively to the photochemical ozone production over the examined metropolitan areas of China, even if during the examined winter months the ozone photochemical production is expected to be minimal. On the other hand, the variations of the background ozone levels within the boundary layer and the free troposphere are not (or very little) discussed.
For this purpose, I think that it would be quite helpful to take into account a relatively recent extended review paper on tropospheric ozone on global scale, including SE Asia which is one of the most important global tropospheric ozone hotspots (Gaudel et al, 2018, Elem Sci Anth, 6: 39. DOI: https://doi.org/10.1525/elementa.291 and also references therein). From my perspective and based on my expertise of analyzing ozone episodes in the Mediterranean region, I would just point out that the possibility of vertical ozone transport in the troposphere influencing the boundary layer and surface ozone values (a major factor in the Mediterranean, especially in its eastern part during summer but also in its western part during spring, is not mentioned in the manuscript and so all measured ozone is considered to be produced by local photochemistry from precursor pollutant emissions emitted in China only. The tropospheric ozone influence on surface ozone concentration is expected to be important during the January – March period when the lockdown measures in China have taken place. In this context, it must be kept in mind that under the examined conditions the back-trajectories are useful mainly for assessing the tropospheric ozone influence in combination to the fact that the synoptic meteorological conditions might lead to high tropospheric, boundary layer and surface ozone concentrations, as indicated in the mentioned suggested references.
Overall, in my opinion, it has to be clearly stated in the manuscript that the photochemical ozone production during the January – March period is quite low, so that the surface ozone variability during the winter lockdown period is essentially determined by the tropospheric and boundary layer ozone background levels in combination with the NO titration in urban and suburban sites, influenced by the NOx emissions. The above remarks must be kept in mind when studying back-trajectories during ozone episodes.
Author Response
Our Revision:
Thanks very much for your review work and outstanding and remarkable suggestions. Your affirmation gives us great encouragement. We will provide a point-by-point response below.
At first, we reduced the text and focused on the description of the main results. The reduced content is listed at the end of the response.
Secondly, we point out the relationship between NO2 and O3 in the part that just mentioned Ox for the first time in the manuscript L204-207. "Since the measurements of NO2 and O3 are interrelated through the rapid titration reaction of NO with O3, producing NO2, especially at urban sites, the comprehensive oxidation capacity (Ox) of the atmosphere can be evaluated by observing O3 and NO2."
Thirdly, we have carefully studied the significant literature (Gaudel et al, 2018, Elem Sci Anth, 6: 39. DOI: https://doi.org/10.1525/elementa.291) and the references therein. These are excellent literature whose results can make up for the lack of analysis in this article. We have absorbed some great contents and introduced important research results of tropospheric ozone into the manuscript.
We revised the manuscript in L178-185. "When compared to January-March the increased ozone in April-June (post lockdown period) were mainly attributed to the seasonal variation of background tropospheric ozone, which determines the ozone variability especially in sites with relatively low primary pollution [26]. And the photochemical ozone production during the January-March period was quite low, so that the surface ozone variability during the winter lockdown period was essentially determined by the tropospheric and boundary layer ozone background levels in combination with the NO titration in urban and suburban sites, influenced by the NOx emissions."
And the revised manuscript L360-368. "Considering the tropospheric ozone influence on surface ozone concentration was expected to be important during the January-March period when the lockdown measures in Shenyang have taken place. As a global major factor in various regions, especially in eastern Mediterranean during summer but also in its western part during spring and many other regions [39-41], the vertical ozone transport in the troposphere influencing the boundary layer and surface ozone values [42,43]. Besides the back-trajectories are useful mainly for assessing the tropospheric ozone influence in combination to the fact that the synoptic meteorological conditions might lead to high tropospheric, boundary layer and surface ozone concentrations [44], it is very necessary to bring the analysis of long-distance transmission to clarify the underlying causes of ozone variation."
And the revised manuscript L477-479. "The tropospheric ozone and its subsequent influence including tropospheric folding and subsidence on the boundary layer and surface ozone concentrations had significant impacts on regional ozone transportation."
[26]Wespes, C, Hurtmans, D, Clerbaux, C, Boynard, A and Coheur, P-F. 2018. Decrease in tropospheric O3 levels of the Northern Hemisphere 1 observed by IASI. Atmos. Chem. Phys. Discuss. DOI: https://doi.org/10.5194/acp-2017-904
[39]Schultz, MG, Schröder, S, Lyapina, O, Cooper, O, Galbally, I, Petropavlovskikh, I, et al. 2017. Tropospheric Ozone Assessment Report: Database and Metrics Data of Global Surface Ozone Observations. Elem Sci Anth. 5: 58. DOI: https://doi.org/10.1525/elementa.244
[40]Fleming, ZL, Doherty, RM, von Schneidemesser, E, Malley, CS, Cooper, OR, Pinto, JP, et al. 2018. Tropospheric Ozone Assessment Report: Presentday ozone distribution and trends relevant to human health. Elem Sci Anth. 6(1): 12. DOI: https://doi.org/10.1525/elementa.273
[41]Young, PJ, Naik, V, et al. 2018. Young, PJ, Naik, V, Fiore, AM, Gaudel, A, Guo, J, Lin, MY, et al. 2018. Tropospheric Ozone Assessment Report: Assessment of global-scale model performance for global and regional ozone distributions, variability, and trends. Elem Sci Anth. 6(1): 10. DOI: https://doi.org/10.1525/elementa.265
[42]Gaudel, A.; Cooper, O.R.; Ancellet, G.; Barret, B.; Boynard, A.; Burrows, J.P.; Clerbaux, C.; Coheur, P.-F.; Cuesta, J.; Cuevas Agulló, E.; et al. Tropospheric Ozone Assessment Report: Present-day distribution and trends of tropospheric ozone relevant to climate and global atmospheric chemistry model evaluation. Elem. Sci. Anthr. 2018, 6, 39. DOI: https://doi.org/10.1525/elementa.291
[43]Steinbrecht, W, et al. 2017. An update on ozone profile trends for the period 2000 to 2016. Atmos. Chem. Phys. 17: 10675–10690. DOI: https://doi.org/10.5194/acp-17-10675-2017
[44]Cohen, Y, et al. 2018. Climatology and long-term evolution of ozone and carbon monoxide in the UTLS at northern mid-latitudes, as seen by IAGOS from 1995 to 2013. Atmos. Chem. Phys., 18: 5415–5453. DOI: https://doi.org/10.5194/acp-18-5415-2018
Reduced content in the manuscript include "According to the data of national open access statistics (http://www.tianditu.gov.cn/index.html), the regional GDP of Liaoning Province (2340.92 billion yuan) is 56.6 and 47.2 percentage points higher than the two neighboring provinces, whose population is 43.746 million", "AQI is calculated from the hourly concentration values of five major pollutants: ground ozone, particulate matter, carbon monoxide, sulfur dioxide and nitrogen dioxide", "the service industry has been closed this year, emissions from various sources have been significantly suppressed", "The CTR station is surrounded by the Hunhe River in the north and Metro Line 9 in the south, where are roughly 20 dense residential areas, two middle schools and several commercial centers nearby", "The ozone concentration was higher in the northeast than southwest before the lockdown, and gradually higher in the northwest than in the southeast after unblocking, while the interesting shift may be related to the direction of the wind", "In the past, the spatial distribution of ozone in the urban center was lower than that in the suburbs was generally motivated by plenty of NO emitted by motor vehicles in the central city consumed O3 and delivered the primary and secondary NO2 to the downwind reacting with VOCs and generated O3, resulting in the increase of ozone concentration in the suburbs", "Prior to this, Russia only closed border ports and international flights, restricted large-scale events with crowds of more than 5,000", " On the one hand, meteorological conditions such as high humidity, low temperature and high wind speed led to low ozone concentration", "In the days of severe ozone pollution, not only low humidity and coordinated temperature changes appeared, but also the extreme weather also promoted the long-distance transmission of pollutants on high pollution paths to have a more serious impact on ozone pollution in Shenyang", " The observation results also showed that the inter-annual variation of PM2.5 and ozone in China was negatively correlated. The precursor components of PM2.5 and ozone overlapped a lot, the relationship between PM2.5 and precursors was mostly linear, but ozone did not", "Although the lockdown measures had different impacts on ozone and nitrogen dioxide, their daily trends were not affected. The peak and valley time of the two were still corresponding to each other, but the occurrence time was delayed, which indicated that the chemical reaction and interaction between pollutants were not affected by lockdown, but the reduction of the concentration of various precursor pollutants caused the reaction time to be delayed, and the time for ozone accumulation and nitrogen dioxide consumption are also delayed", "high-concentration ozone in Shenyang was distributed in the northeast, while low-concentration ozone was still in the central urban area", "It was found that the long-distance transport of high-concentration ozone had a greater impact on the ozone pollution in Shenyang in the same period of time when China's surrounding countries and regions such as Russia and Mongolia did not take lockdown measures", "Compared with the rapid decline in the concentration of other pollutants, ozone has increased uncharacteristically. Those comprehensive factors indicated that due to the substantial reduction in precursor contaminants and the accompanying reduction in titration effects, the atmospheric environment showed a decrease in the concentration of accumulated nitrogen dioxide, and the total ozone concentration increased sharply, whose peak time was delayed and concentration was higher".
Submission Date
10 November 2020
Date of this review
20 Nov 2020 12:48:40

Round 2
Reviewer 3 Report
I think that the authors have responded sufficiently to my comments and the paper could be published in principle, after a careful language check. I would also suggest making a last check to verify that the inserted changes are in line with other presented arguments in the paper.